# EDEM2 stably disulfide-bonded to TXNDC11 catalyzes the first mannose trimming step in mammalian glycoprotein ERAD

Ginto George[1†], Satoshi Ninagawa[1†], Hirokazu Yagi[2], Taiki Saito[2], Tokiro Ishikawa[1], Tetsushi Sakuma[3], Takashi Yamamoto[3], Koshi Imami[4], Yasushi Ishihama[4], Koichi Kato[2,5], Tetsuya Okada[1*], Kazutoshi Mori[1*]

[1]Department of Biophysics, Graduate School of Science, Kyoto University, Kyoto, Japan; [2]Graduate School of Pharmaceutical Sciences, Nagoya City University, Nagoya, Japan; [3]Division of Integrated Sciences for Life, Graduate School of Integrated Sciences for Life, Hiroshima University, Hiroshima, Japan; [4]Department of Molecular and Cellular BioAnalysis, Graduate School of Pharmaceutical Sciences, Kyoto University, Kyoto, Japan; [5]Exploratory Research Center on Life and Living Systems (ExCELLS) and Institute for Molecular Science, National Institutes of Natural Sciences, Okazaki, Japan

**Abstract** Sequential mannose trimming of N-glycan ($Man_9GlcNAc_2$ -> $Man_8GlcNAc_2$ -> $Man_7GlcNAc_2$) facilitates endoplasmic reticulum-associated degradation of misfolded glycoproteins (gpERAD). Our gene knockout experiments in human HCT116 cells have revealed that EDEM2 is required for the first step. However, it was previously shown that purified EDEM2 exhibited no $\alpha$1,2-mannosidase activity toward $Man_9GlcNAc_2$ in vitro. Here, we found that EDEM2 was stably disulfide-bonded to TXNDC11, an endoplasmic reticulum protein containing five thioredoxin (Trx)-like domains. C558 present outside of the mannosidase homology domain of EDEM2 was linked to C692 in Trx5, which solely contains the CXXC motif in TXNDC11. This covalent bonding was essential for mannose trimming and subsequent gpERAD in HCT116 cells. Furthermore, EDEM2-TXNDC11 complex purified from transfected HCT116 cells converted $Man_9GlcNAc_2$ to $Man_8GlcNAc_2$(isomerB) in vitro. Our results establish the role of EDEM2 as an initiator of gpERAD, and represent the first clear demonstration of in vitro mannosidase activity of EDEM family proteins.

*For correspondence:
tokada@upr.biophys.kyoto-u.ac.jp (TO);
mori@upr.biophys.kyoto-u.ac.jp (KM)

†These authors contributed equally to this work

Competing interests: The authors declare that no competing interests exist.

## Introduction

Failure in protein folding in the endoplasmic reticulum (ER) detrimentally affects the maturation of secretory and transmembrane proteins, which comprise one-third of total proteins synthesized in mammalian cells. As misfolded proteins are not only unable to fulfill the functions assigned them by genetic code but may also exert proteotoxicity by making inappropriate interactions with other functional proteins, protein misfolding constitutes a fundamental threat to all living cells. Proteins misfolded in the lumen of the ER are recognized, delivered to and translocated through the multiprotein complex termed the retrotranslocon in the ER membrane, ubiquitinated, and degraded by the proteasome in the cytosol, a series of events collectively termed ER-associated degradation (ERAD), and specifically termed ERAD-L (L for lumen). Mammalian ERAD-L deals with the degradation of both non-glycoproteins (non-gpERAD) and glycoproteins (gpERAD), in which the carbohydrate unit $Glc_3Man_9GlcNAc_2$ (see *Figure 1A*) is covalently attached to the asparagine residue

**Figure 1.** Effect of mutation of various cysteine residues in EDEM2 on gpERAD. (**A**) Structure of Glc$_3$Man$_9$GlcNAc$_2$ is schematically shown. Mannose residues are α1,2-bonded or α1,6-bonded as indicated. (**B**) Structures of human EDEM2 and yeast Htm1p are schematically shown with cysteine residues (C) highlighted together with their positions (black bars underneath C indicate conserved cysteine residues, whereas white bars over C indicate non-conserved cysteine residues). The purple and yellow boxes denote the signal sequence and mannosidase homology domain (MHD), respectively.

*Figure 1 continued on next page*

Figure 1 continued

Sequence comparison around the four cysteine residues conserved between EDEM2 and Htm1p (marked in red color) is shown below (asterisk and colon indicate identical and similar amino acids, respectively). (C) Cell lysates were prepared from WT and EDEM2-KO cells expressing mCD3-δ-ΔTM-HA by transfection, treated with (+) or without (-) EndoH, and analyzed by immunoblotting using anti-HA antibody. mCD3-δ-ΔTM-HA* denotes deglycosylated mCD3-δ-ΔTM-HA. (D) Cell lysates were prepared from WT and EDEM2-KO cells expressing WT or one of various cysteine mutants of 3x Flag-tagged EDEM2 together with mCD3-δ-ΔTM-HA by transfection, and analyzed by immunoblotting using anti-HA and anti-EDEM2 antibodies. E117Q is an enzymatically inactive mutant of EDEM2. (E) Cycloheximide chase was conducted to determine the degradation rate of mCD3-δ-ΔTM-HA in WT and EDEM2-KO cells expressing WT or one of the three cysteine mutants of 3x Flag-tagged EDEM2 by transfection, and analyzed by immunoblotting using anti-HA antibody (n = 3). Quantified data are shown on the right.

present in the triplet code (Asn-X-Ser/Thr; X: any amino acid except Pro) for N-glycosylation. ERAD-L is further categorized into ERAD-Ls (for soluble proteins) and ERAD-Lm (for membrane proteins) (**Brodsky, 2012**; **Smith et al., 2011**; **Xie and Ng, 2010**).

After the conversion of $Glc_3Man_9GlcNAc_2$ to $Man_9GlcNAc_2$ through trimming of three glucose residues in the A chain by two enzymes, glucosidase I and glucosidase II, the number and position of the remaining mannose residues becomes a key determinant of the efficacy of gpERAD (see **Figure 1A**) (**Kamiya et al., 2012**). We have conducted genetic (knockout) analyses of four α1,2-mannosidase candidate genes able to trim mannose residues in the ER, namely ER mannosidase I (ERmanI), EDEM1, EDEM2 and EDEM3 (**Ninagawa et al., 2014**), using both conventional homologous recombination in chicken DT40 cells and transcription activator-like effector nuclease-mediated genome editing in human HCT116 diploid cells (**Roschke et al., 2002**). Results clearly showed that EDEM2 is required for conversion of $Man_9GlcNAc_2$ (M9 hereafter) to $Man_8GlcNAc_2$ (M8B hereafter; note that only the outermost mannose of the B branch is removed); and that EDEM3 and EDEM1 are required for conversion of M8B to $Man_7GlcNAc_2$ (M7A hereafter; note that only the A branch has the outermost mannose), $Man_6GlcNAc_2$ and $Man_5GlcNAc_2$. These oligosaccharides are then recognized by specific lectin molecules such as OS9 and XTP3B for delivery to the retrotranslocon (**van der Goot et al., 2018**), because the α1,6-mannose linkage becomes exposed (see **Figure 1A**). ERmanI appears to trim the outermost mannose of the B branch randomly regardless of the number of mannose residues rather than specifically convert M9 to M8B (**Ninagawa et al., 2014**).

Nonetheless, given previous findings that His-tagged EDEM2 purified or HA-tagged EDEM2 immunoprecipitated from transfected HEK293 cells exhibited little or no mannosidase activity toward pyridylamine (PA)-tagged M9 or PA-tagged M8 in vitro (**Mast et al., 2005**; **Shenkman et al., 2018**), EDEM2's α1,2-mannosidase activity remains to be demonstrated in vitro. Interestingly, Htm1p, a yeast homolog of mammalian EDEM1/2/3, is physically and stably associated with Pdi1p (protein disulfide isomerase) (**Clerc et al., 2009**; **Sakoh-Nakatogawa et al., 2009**). By forming an intermolecular disulfide bond with Cys579 or Cys644 of Htm1p (shown in red in **Figure 1B**), Pdi1p facilitates intramolecular disulfide bond formation between Cys65 and Cys445 of Htm1p (shown in blue in **Figure 1B**), which is critical for Htm1p to function in gpERAD (**Sakoh-Nakatogawa et al., 2009**). Importantly, Htm1p-Pdi1p complex purified from insect cells, in which glutathione S-transferase-Htm1p fusion protein and His-tagged Pdi1p are simultaneously expressed, converted approximately 10% of M8 to M7 in vitro, only when M8 is attached to unfolded, reduced and alkylated polypeptides; in contrast, Htm1p-Pdi1p complex was nearly inactive against free M8 (**Gauss et al., 2011**). Therefore, by analogy, EDEM2 may require a PDI-like partner protein to exhibit its mannosidase activity.

Recently, comprehensive genetic analyses were conducted in human haploid cells by way of conventional gene-trap mutagenesis or CRISPR/Cas9-mediated gene knockout to identify genes involved in ERAD of a glycoprotein termed major histocompatibility complex (MHC) class I molecule (**Timms et al., 2016**). Both approaches identified EDEM2 and TXNDC11, a type II transmembrane protein containing five thioredoxin (Trx)-like domains (see **Figure 2C**). TXNDC11 was shown to be N-glycosylated, localized in the ER and required for gpERAD of various substrates, including CD3δ, TCRα and NHK (null Hong Kong variant of α1-proteinase inhibitor) in addition to MHC class I, but not for NHK-QQQ, a non-glycosylated version of NHK. TXNDC11 in the gpERAD assay was inactivated by the double mutation of Cys692 and Cys695 in the CxxC motif of the Trx5 domain, but not by the mutation of Cys137 in the CxxS motif of the Trx1 domain. The Trx5 domain expressed and purified from *E. coli* cells exhibited reductase activity in vitro. Immunoprecipitation followed by mass

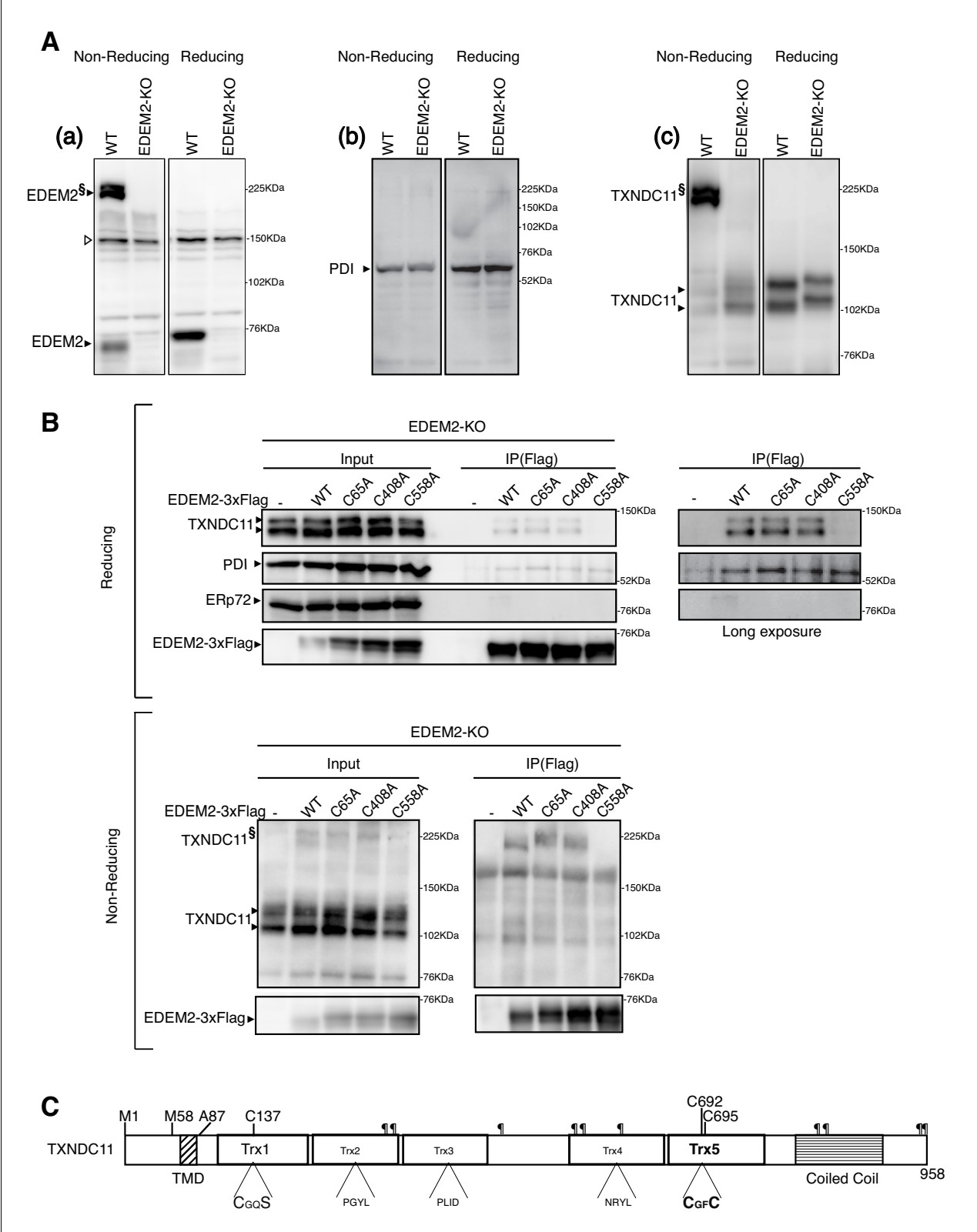

**Figure 2.** Disulfide bond formation between EDEM2 and TXNDC11. (A) Cell lysates were prepared from WT and EDEM2-KO cells, subjected to SDS-PAGE under reducing and non-reducing conditions, and analyzed by immunoblotting using anti-EDEM2 (a), anti-PDI (b) and anti-TXNDC11 (c) antibodies. § denotes high molecular weight forms of EDEM2 and TXNDC11. Open triangle indicates a non-specific band. (B) Cell lysates were prepared from EDEM2-KO cells expressing WT or one of the three cysteine mutants of 3x Flag-tagged EDEM2 by transfection, and subjected to

*Figure 2 continued on next page*

*Figure 2 continued*

immunoprecipitation using anti-Flag antibody. An aliquot of cell lysates (Input) and immunoprecipitates {IP(Flag)} were subjected to SDS-PAGE under reducing and non-reducing conditions, and analyzed by immunoblotting using anti-TXNDC11, anti-PDI, anti-ERp72, and anti-Flag antibodies. (**C**) Structure of human TXNDC11 containing the TMD, five Trx domains, and coiled coil domain is schematically shown. ¶ denote potential N-glycosylation sites. The positions of two initiation methionines are also shown.

spectrometric analysis revealed that TXNDC11 bound to PDI, EDEM2, GANAB, EDEM3, GLU2B and TXNDC5 (*Timms et al., 2016*).

Here, we demonstrated that EDEM2 is stably disulfide bonded to TXNDC11 and that the purified EDEM2-TXNDC11 complex is capable of converting PA-M9 to PA-M8B in vitro.

## Results

### EDEM2 is disulfide-bonded to TXNDC11

Human EDEM2 contains a total of eight cysteine residues, among which four are localized in regions conserved with yeast Htm1p (*Figure 1B*, shown with black bars). We mutated each cysteine residue of EDEM2 to alanine and examined the resulting effect on degradation of the ERAD-Ls substrate mCD3-δ-ΔTM-HA containing three N-glycosylation sites (*Bernasconi et al., 2010*). mCD3-δ-ΔTM-HA migrated more slowly in EDEM2-KO cells than in WT cells due to the absence of the first mannose trimming activity (M9 -> M8B) in EDEM2-KO cells; as expected, this migration difference was lost after treatment with endoglycosidase H (EndoH) (*Figure 1C*). Introduction of 3x Flag-tagged WT EDEM2 into EDEM2-KO cells restored the mannose trimming activity, but introduction of three of the eight cysteine mutants (C65A, C408A and C558A) did not do so (*Figure 1D*), similarly to the catalytically inactive E117Q mutant of EDEM2 (*Ninagawa et al., 2014*). Cycloheximide chase experiments showed that mCD3-δ-ΔTM-HA was rapidly degraded in WT cells but not in EDEM2-KO cells. Introduction of WT EDEM2 but not the three cysteine mutants into EDEM2-KO cells restored this degradation activity (*Figure 1E*).

We noticed that EDEM2 was detected as both monomer and high molecular weight forms (a doublet band) when analyzed by non-reducing SDS-PAGE (*Figure 2Aa*). The high molecular weight forms did not react with anti-PDI antibody (*Figure 2Ab*) but appeared to react with anti-TXNDC11 antibody (*Figure 2Ac*), suggesting that EDEM2 is disulfide-bonded to TXNDC11 (see below for the reasons why monomer TXNDC11 was detected as a doublet band and why they migrated more slowly in EDEM2-KO cells than in WT cells). To determine which cysteine residue of EDEM2 is involved in the presumed covalent association with TXNDC11, immunoprecipitation using anti-Flag antibody was carried out in EDEM2-KO cells into which 3x Flag-tagged WT EDEM2 or one of the three cysteine mutants had been transfected. Results showed that endogenous TXNDC11 was co-immunoprecipitated with WT, C65A and C408A but not with C558A EDEM2 (*Figure 2B*, Reducing). In contrast, endogenous PDI was co-immunoprecipitated with all four EDEM2, whereas endogenous ERp72 was not co-immunoprecipitated with any of them (*Figure 2B*, Reducing). When analyzed under non-reducing conditions, high molecular weight forms of TXNDC11 disappeared in immunoprecipitates from cells expressing C558A EDEM2 (*Figure 2B*, Non-reducing). These results clearly indicated that EDEM2 is disulfide-bonded to TXNDC11 through its cysteine 558 residue.

It should be noted that TXNDC11 contains from its N-terminus a single transmembrane domain (TMD), five Trx-like domains, and a coiled coil domain, as well as ten potential N-glycosylation sites (*Figure 2C*) (*Timms et al., 2016*).

### TXNDC11 knockout eliminates high molecular weight forms of EDEM2 required for gpERAD

We intended to knock out the *TXNDC11* gene in HCT116 diploid cells using the CRISPR/Cas9-based Precise Integration into Target Chromosome (PITCh) method and targeted its exon one with the puromycin-resistant gene flanked by the left and right arms of the *TXNDC11* gene as well as guide RNA target sites (*Figure 3—figure supplement 1A*). We analyzed 34 puromycin-resistant colonies by genomic PCR and found that two independent clones (KO#1 and KO#2) had an insertion whose size was comparable to that of the puromycin-resistant gene. However, sequencing of the genomic

PCR products (see *Figure 3—figure supplement 1D*) indicated that Cas9 cleaved the targeting vector on only one side (upstream of the left arm in the case of KO#1 or downstream of the right arm in the case of KO#2), and therefore the whole targeting plasmid was integrated into the exon one in one allele of KO#1 or KO#2 cells. Interestingly, a fragment(s) of the targeting vector or Cas9-expressing vector was inserted into exon one in the other allele of KO#1 or KO#2 cells (Fr. A and B in KO#1; Fr. C in KO#2) (*Figure 3—figure supplement 1B and C*), which inactivated the expression of TXNDC11 at both mRNA and protein levels (*Figure 3—figure supplement 1E and F*).

Quantitative RT-PCR showed that the level of *EDEM2* mRNA was not significantly affected by TXNDC11 knockout (*Figure 3Aa*). The level of spliced *XBP1* mRNA was also not affected (*Figure 3Ab*), indicating that TXNDC11-KO cells are not significantly ER-stressed. Importantly, immunoblotting showed that high molecular weight forms of EDEM2 were lost in both TXNDC11-KO#1 and -KO#2 cells, as we expected (*Figure 3B*). Introduction of 3x Flag-tagged TXNDC11 alone into TXNDC11-KO#1 cells produced monomer and probably aggregated TXNDC11, whereas simultaneous introduction of Myc-tagged EDEM2 and 3x Flag-tagged TXNDC11 produced a high molecular weight complex of EDEM2 and TXNDC11 (*Figure 3C*). High molecular weight forms of EDEM2 present in WT cells were more resistant to trypsin digestion than monomer EDEM2 present in TXNDC11-KO#1 cells (*Figure 3D*), indicative of their differential conformations in the presence or absence of the partner protein TXNDC11.

Degradation of endogenous ATF6α, an ERAD-Lm substrate whose degradation is initiated by EDEM2-mediated first mannose trimming (*Ninagawa et al., 2014*), was markedly delayed in TXNDC11-KO#1 and -KO#2 cells, as in EDEM2-KO cells, compared with WT cells (*Figure 4A*). Similarly, degradation of mCD3-δ-ΔTM-HA was delayed in both TXNDC11-KO#1 and -KO#2 cells, compared with WT cells (*Figure 4B*), which was consistent with the migration position of mCD3-δ-ΔTM-HA in reducing SDS-PAGE (*Figure 4C*). N-glycan profiling revealed that oligosaccharide M9 was accumulated in TXNDC11-KO#1 and -KO#2 cells, as in the case of EDEM2-KO cells (*Figure 4D and E*); accumulation of $Glc_1Man_9GlcNAc_2$ (GM9) was probably due to a subtle change in culture conditions, because GM9 was not accumulated in EDEM2-KO human HCT116 cells in our previous study (*Ninagawa et al., 2014*). Alternatively, it may be due to enhanced cellular re-glucosylation activity caused by blockage of mannose trimming from M9 to M8 (*Molinari, 2007*), because both GM9 and M9 were accumulated in EDEM2-KO chicken DT40 cells in our previous study (*Ninagawa et al., 2014*). Introduction of 3x Flag-tagged TXNDC11 (WT) into TXNDC11-KO#1 or -KO#2 cells restored mannose trimming activity, as expected (*Figure 4C*). These results clearly showed that high molecular weight forms of EDEM2 complexed with TXNDC11 but not monomer EDEM2 possess catalytic mannosidase activity. Hereafter, TXNDC11-KO#1 cells are used as TXNDC11-KO cells.

Immunoblotting showed that both endogenous and transfected (WT) TXNDC11 was consistently detected as a doublet band in SDS-PAGE (*Figures 2*, *3* and *5A*). EndoH treatment revealed that both forms were N-glycosylated (*Figure 5A*), indicating that the doublet band migrated more slowly in EDEM2-KO cells than in WT cells due to the absence of the first mannose trimming activity in EDEM2-KO cells (see *Figure 2Ac*); indeed, the migration difference of TXNDC11 between WT and EDEM2-KO cells was lost after EndoH treatment (data not shown). Because TXNDC11 has two methionine residues (M1 and M58) at the N-terminus of the TMD (*Figure 2C*), and because the sequences around both M1 (catGtaATGt) and M58 (ttcctcATGG) resemble the Kozak consensus sequence for translational initiation (gccRccATGG), we sought the possibility that translation started at two methionine residues. We therefore mutated M1 and M58 to alanine and introduced them into TXNDC11-KO cells separately. Immunoblotting revealed that the M1A and M58A mutants showed only faster and slower migrating bands, respectively, as we expected, and both were N-glycosylated (*Figure 5A*), although we have no idea why the M58A mutant was still detected as a doublet band in a previous report (*Timms et al., 2016*). Both M1A and M58A of TXNDC11 are functional in mannose trimming activity, as evidenced by the difference in migration position of mCD3-δ-ΔTM-HA in TXNDC11-KO cells expressing WT, M1A, and M58A TXNDC11, compared with untransfected TXNDC11-KO cells (*Figure 5B*).

Centrifugal fractionation after repeated freezing and thawing of cells indicated that the M58A mutant was a membrane protein like calnexin, whereas the M1A mutant was split into a calnexin-like membrane protein and a calreticulin-like soluble protein (*Figure 5C*). This suggests that the TMD acted as a relatively weak signal peptide for M1A. SignalP-5.0 (http://www.cbs.dtu.dk/services/SignalP/) predicted that the probability for functionality of the TMD as a signal peptide is between 0.6

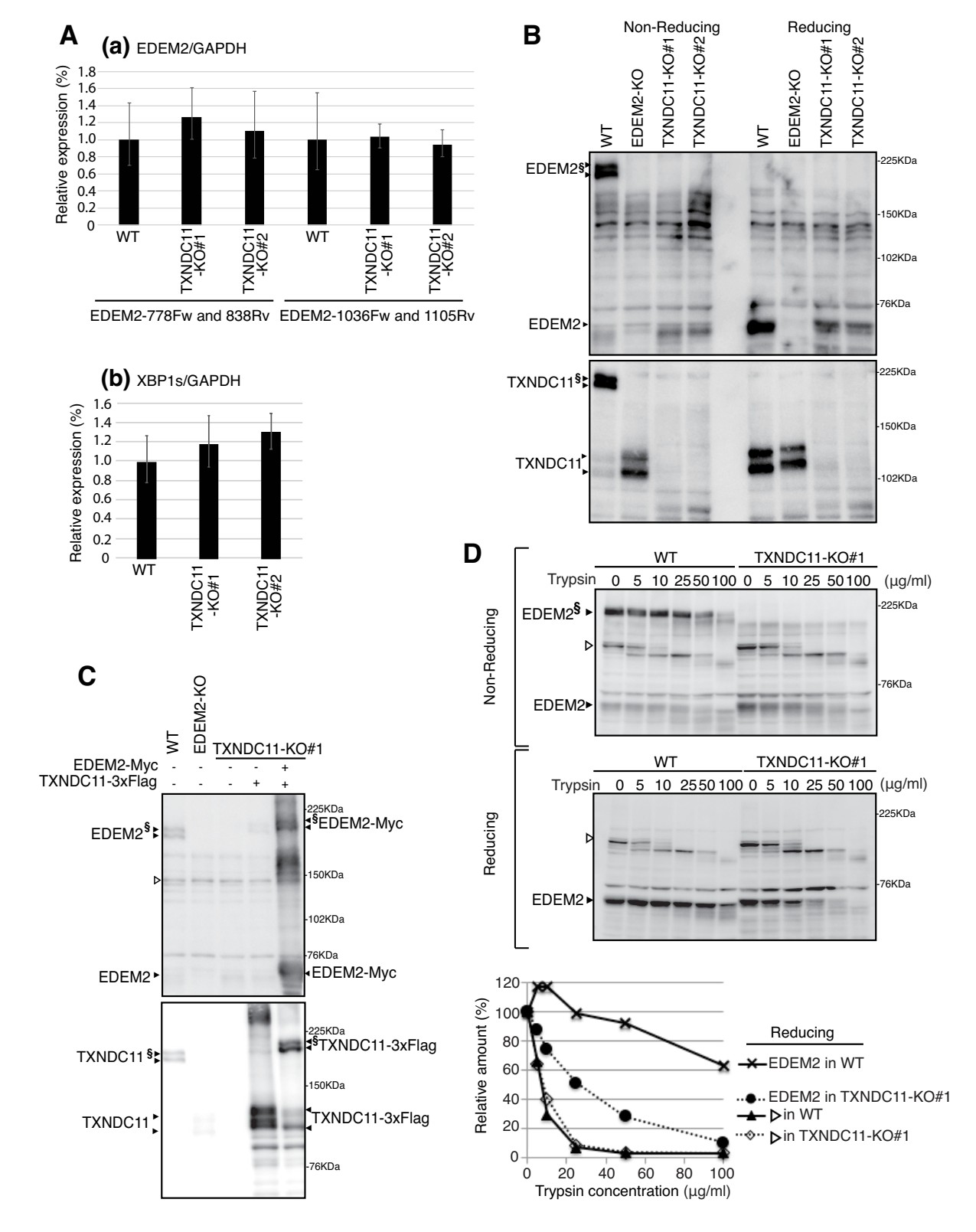

**Figure 3.** Effect of TXNDC11 knockout on EDEM2. (**A**) Quantitative RT-PCR was conducted to determine the levels of endogenous *EDEM2* mRNA (a) using the two primer sets indicated as well as spliced *XBP1* mRNA (b) relative to the level of *GAPDH* mRNA in WT and two TXNDC11-KO cells (n = 3). (**B**) Cell lysates were prepared from WT, EDEM2-KO, and two TXNDC11-KO cells, subjected to SDS-PAGE under reducing and non-reducing conditions, and analyzed by immunoblotting using anti-EDEM2 and anti-TXNDC11 antibodies. (**C**) Cell lysates were prepared from WT, EDEM2-KO,

*Figure 3 continued on next page*

*Figure 3 continued*

and TXNDC11-KO#1 cells expressing 3x Flag-tagged TXNDC11 or both 3x Flag-tagged TXNDC11 and Myc-tagged EDEM2 by transfection, subjected to SDS-PAGE under non-reducing conditions, and analyzed by immunoblotting using anti-EDEM2 and anti-TXNDC11 antibodies. (D) Cell lysates were prepared from WT and TXNDC11-KO#1 cells, treated with the indicated amount of trypsin at 4°C for 15 min, subjected to SDS-PAGE under reducing and non-reducing conditions, and analyzed by immunoblotting using anti-EDEM2 antibody. The band with open triangle denotes a non-specific protein which serves as a control for trypsin digestion. Quantified data are shown at the bottom.

The online version of this article includes the following figure supplement(s) for figure 3:

**Figure supplement 1.** Construction of TXNDC11-KO cells.

and 0.7 and that the probability for its cleavage at the C-terminus of A87 is approximately 0.5 (*Figure 5D*). We thus constructed the ΔSP (lacking amino acids from A59 to A87, see *Figure 2C*) and A87F (having the probability for functionality of the TMD as a signal peptide and the probability for its cleavage are less than 0.4 and approximately 0.1, respectively, *Figure 5D*) mutants in the M1A mutant and introduced them into TXNDC11-KO cells separately. M1A-ΔSP was synthesized as an unglycosylated cytosolic protein, as expected (*Figure 5E*). M1A-A87F moved slightly slower than M1A before and after EndoH treatment during reducing SDS-PAGE (*Figure 5E*) and M1A-A87F was mostly a calnexin-like membrane protein (*Figure 5F*). We concluded that M1A becomes a soluble protein if it is cleaved by signal peptidase and that M1A uncleaved by chance remains as a M58A-like membrane protein.

## EDEM2 stably disulfide-bonded to TXNDC11 has mannosidase activity

TXNDC11 contains a total of 24 cysteine residues (*Figure 6A*). To determine which cysteine residue of TXNDC11 is involved in disulfide-bonding with EDEM2, we mutated each cysteine to serine (except for C5 located at the N-terminal end) and transfected TXNDC11-KO cells together with Myc-tagged EDEM2. Analysis by immunoblotting after non-reducing SDS-PAGE revealed that high molecular weight forms were lost in cells expressing Myc-tagged EDEM2 and 3x Flag-tagged C692S mutant of TXNDC11 (*Figure 6B*). This observation was confirmed by immunoprecipitation in cells expressing WT or one of the three cysteine mutants of 3x Flag-tagged TXNDC11 (*Figure 6C*). The C692S mutant did not restore mannose trimming activity (*Figure 6D*) and therefore gpERAD functionality (*Figure 6E*) in TXNDC11-KO cells, in contrast to C137S or C695S mutant of TXNDC11.

We intended to purify EDEM2 complexed with TXNDC11 and uncomplexed EDEM2 to determine their mannosidase activity in vitro. To this end, EDEM2-KO cells were untransfected or transfected with plasmid to express TXNDC11(M1A)-3xFlag, a soluble protein at least in part, and plasmid to express WT or C558A mutant of tandem affinity purification (TAP)-tagged EDEM2; this TAP consists of 2x Immunoglobulin G-binding site of protein A, 2x TEV protease recognition site, and 6x Myc. TAP-EDEM2 was purified using IgG Sepharose beads, from which 6xMyc-EDEM2 was eluted by digestion with TEV protease. Silver staining of the eluates after reducing SDS-PAGE revealed the expected bands. TXNDC11(M1A)-3xFlag was purified with WT but not with C558A EDEM2 (*Figure 7A*). Given that a marginal amount of TXNDC11(M1A)-3xFlag was detected in the eluate for C558A EDEM2 (*Figure 7B*), TXNDC11 might physically associate with EDEM2 even in the absence of disulfide bonding, albeit only slightly.

Equal amounts of 6xMyc-EDEM2 (checked by immunoblotting, *Figure 7B*) uncomplexed or complexed with TXNDC11(M1A)-3xFlag were subjected to in vitro mannosidase assay. Results showed that PA-M9 was converted to PA-M8 by EDEM2 complexed with TXNDC11 but not by uncomplexed (C558A) EDEM2 (*Figure 7C*) in an incubation period-dependent manner. This M8 peak was determined to be M8B (*Figure 7D*).

## Discussion

Mannose trimming initiates gpERAD. In this paper, we intended to fill the huge gap which has arisen between genetic and biochemical analyses of EDEM2. Our gene knockout analysis clearly showed that EDEM2 is required for conversion of M9 to M8B (*Ninagawa et al., 2014*), but purified EDEM2 shows no mannosidase activity toward free (not bound to protein) M9 oligosaccharide in vitro (*Mast et al., 2005*; *Shenkman et al., 2018*). Moreman and colleagues used EDEM2 purified using a nickel column from HEK293 cells stably overexpressing His-tagged EDEM2 by transfection, and

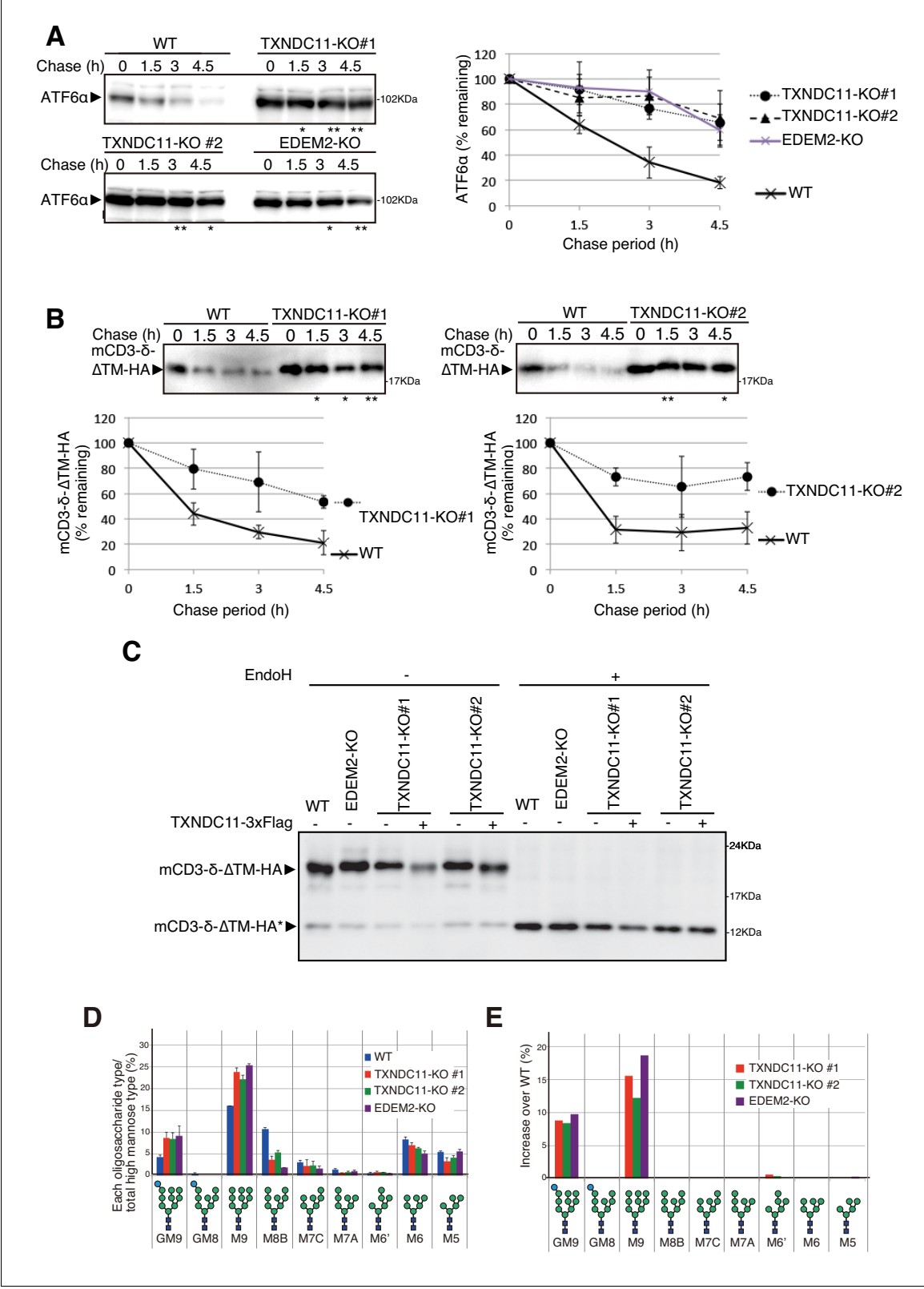

**Figure 4.** Effect of TXNDC11 knockout on gpERAD. (**A**) Cycloheximide chase was conducted to determine the degradation rate of endogenous ATF6α in WT, EDEM2-KO and two TXNDC11-KO cells, and cell lysates were analyzed by immunoblotting using anti-ATF6α antibody (n = 3). Quantified data are shown on the right. (**B**) Cycloheximide chase was conducted to determine the degradation rate of mCD3-δ-ΔTM-HA in transfected WT and two TXNDC11-KO cells, and cell lysates were analyzed by immunoblotting using anti-HA antibody (n = 3). Quantified data are shown below. (**C**) WT and

*Figure 4 continued on next page*

*Figure 4 continued*

EDEM2-KO cells were transfected with the plasmid to express mCD3-δ-ΔTM-HA. The two TXNDC11-KO cells were also transfected with the plasmid to express mCD3-δ-ΔTM-HA together with (+) or without (-) the plasmid to express 3x Flag-tagged TXNDC11(WT). Cell lysates were then prepared, treated with (+) or without (-) EndoH, and analyzed by immunoblotting using anti-HA antibody. (**D**) Isomer composition of N-glycans prepared from total cellular glycoproteins of WT, TXNDC11-KO#1, TXNDC11-KO#2 and EDEM2-KO cells is shown. This experiment was completed once. (**E**) Oligosaccharides obtained in (**D**) whose contents in TXNDC11-KO#1, TXNDC11-KO#2 and EDEM2-KO cells exceeded those in WT cells are displayed with increase over WT (%).

found that their EDEM2 did not convert M9 to M8 in vitro, in marked contrast to purified ERmanI (*Mast et al., 2005*). Lederkremer and colleagues used EDEM2 immunoprecipitated using anti-HA antibody from HEK293 cells overexpressing HA-tagged EDEM2 by transfection, and found that their EDEM2 also did not convert M9 to M8 in vitro, again in marked contrast to purified ERmanI (*Shenkman et al., 2018*).

The key to filling the gap turned out to be complex formation. We found that EDEM2 was stably disulfide-bonded to TXNDC11 through C558 of EDEM2 and C692 of TXNDC11. This stable linkage is essential for EDEM2 to convert M9 to M8B in vitro (*Figure 7*). Accordingly, mannose trimming activity was lost and degradation of the ERAD-Ls substrate mCD3-δ-ΔTM-HA was delayed in EDEM2-KO cells expressing the C558A mutant of EDEM2 (*Figure 1*), and in TXNDC11-KO cells (*Figure 4*) and TXNDC11-KO cells expressing the C692S mutant of TXNDC11 (*Figure 6*). Immunoblotting showed that a majority of TXNDC11 is disulfide-bonded to EDEM2 in WT cells, which alters conformation of EDEM2 to perhaps a better structure for the expression of mannosidase activity (*Figure 3*).

Among the three cysteine residues (C65, C408 and C558) of EDEM2 required for mannose trimming activity and degradation of mCD3-δ-ΔTM-HA (*Figure 1*), C65A and C408A are present in the mannosidase homology domain and likely to be disulfide-bonded intramolecularly, similarly to C65 and C445 in yeast Htm1p, as the disulfide bonding between C65 and C445 is required for the mannosidase activity of Htm1p (*Sakoh-Nakatogawa et al., 2009*). Because EDEM2 associates with PDI (*Figure 2*), disulfide bonding between C65 and C408 is likely to be catalyzed by PDI.

C65 and C408 of EDEM2 are conserved as C83 and C442 of EDEM3. Interestingly, it was recently shown that these cysteine residues in EDEM3 are disulfide-bonded to the CXXC motif in ERp46 containing three Trx domains to form a high molecular weight complex (*Yu et al., 2018*); ERp46 is an oxidoreductase and an efficient disulfide bond introducer in the ER (*Kojima et al., 2014*). This covalent bonding is perhaps transient prior to the formation of disulfide bonding between C83 and C442 of EDEM3, which is required for its mannosidase activity, because either C83S mutation or C442S mutation eliminates such high molecular weight complexes (*Yu et al., 2018*). Nonetheless, although coexpression of EDEM3 and ERp46 appeared to enhance EDEM3 activity in gpERAD assay, it has not yet been shown that ERp46 is required for the mannosidase activity of EDEM3 in vivo and in vitro.

In marked contrast, EDEM2 is stably disulfide-bonded to TXNDC11 through C558 outside of the mannosidase homology domain, and only one cysteine residue of the CXXC motif (C692) of TXNDC11 is used for such covalent bonding. This can be explained by the finding that the C692-containing Trx5 domain of TXNDC11 purified from *E. coli* cells exhibited reduction potential as a reductase rather than an oxidase (*Timms et al., 2016*). C558 of EDEM2 may be disulfide-bonded to C488 (or C464 or C487) (see *Figure 1*) by the action of PDI and then reduced by the Trx5 domain of TXNDC11, but the reductase activity of TXNDC11 is not strong enough for complete reduction of the disulfide bond, leading to the formation of stable disulfide bonding between C558 of EDEM2 and C692 of TXNDC11.

Lederkremer and colleagues also noticed that EDEM2 associates with TXNDC11. They immuno-precipitated EDEM2 using anti-HA antibody from HEK293 cells overexpressing HA-tagged EDEM2 as well as Flag-tagged TXNDC11 by transfection, but failed to show EDEM2's mannosidase activity in vitro (*Shenkman et al., 2018*), in contrast to our purification. Here is to be found the trick: TXNDC11 is always detected as a doublet band due to alternative translational initiation at either M1 or M58, and the M1A mutant protein is soluble at least in part while the M58A mutant protein is transmembrane (*Figure 5*). These researchers appeared to transfect HEK293 cells with HA-tagged EDEM2 cDNA (described as EDEM2-HA) and full-length TXNDC11 cDNA tagged with Flag at the

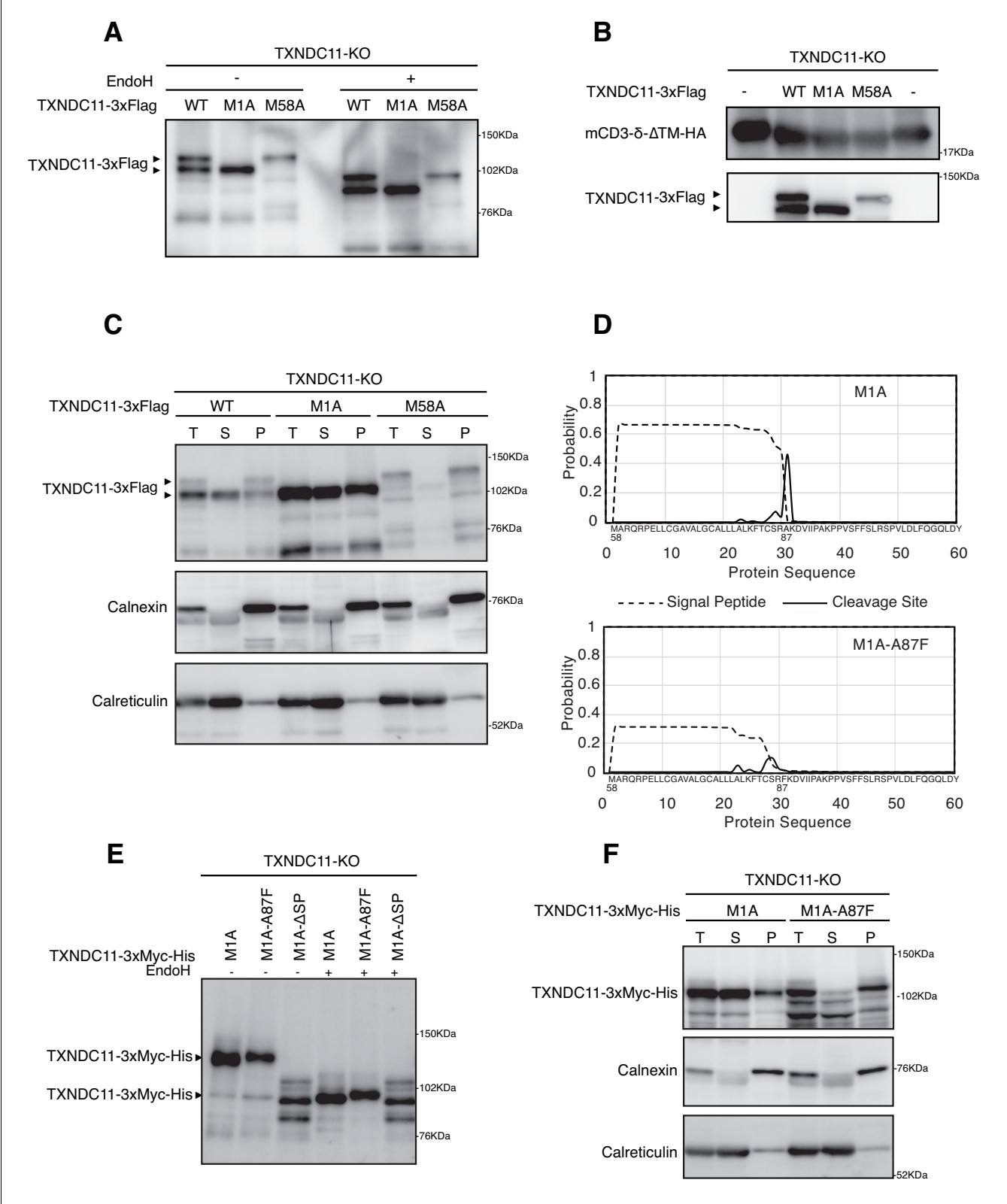

**Figure 5.** Effect of alternative translation on solubility of TXNDC11. (**A**) Cell lysates were prepared from TXNDC11-KO cells expressing WT, M1A or M58A mutant of 3x Flag-tagged TXNDC11 by transfection, treated with (+) or without (-) EndoH, and analyzed by immunoblotting using anti-Flag antibody. (**B**) Cell lysates were prepared from TXNDC11-KO cells expressing mCD3-δ-ΔTM-HA together with WT, M1A, or M58A mutant of 3x Flag-tagged TXNDC11 by transfection, and analyzed by immunoblotting using anti-TXNDC11 and anti-HA antibodies. (**C**) TXNDC11-KO cells expressing
*Figure 5 continued on next page*

*Figure 5 continued*

WT, M1A or M58A mutant of 3x Flag-tagged TXNDC11 by transfection were subjected to repeated freezing and thawing, and then centrifuged as described in Materials and methods. An aliquot of total membrane fraction (T) and resulting supernatant (S) and pellet (P) were analyzed by immunoblotting using anti-TXNDC11, anti-calnexin and anti-calreticulin antibodies. (D) SignalP-5.0-mediated prediction of the probability for functionality of the TMD as a signal peptide (broken line) and the probability for its cleavage by signal peptidase (solid line) in M1A (upper) and M1A-A87F (lower). (E) Cell lysates were prepared from TXNDC11-KO cells expressing M1A, M1A-A87F, or M1A-ΔSP mutant of 3x Myc-His-tagged TXNDC11 by transfection, treated with (+) or without (-) EndoH, and analyzed by immunoblotting using anti-TXNDC11 antibody. (F) TXNDC11-KO cells expressing M1A or M1A-A87F mutant of 3x Myc-His-tagged TXNDC11 by transfection were analyzed as in (C).

N-terminus (described as Flag-TXNDC11). Accordingly, translation was initiated from the methionine located immediately upstream of the Flag tag, resulting in detection of TXNDC11 as a single band, namely as a transmembrane protein, similar to the M58A mutant protein. Therefore, they might not have been able to obtain sufficient EDEM2-TXNDC11 complex in supernatant after lysis. In contrast, we transfected HCT116 cells with TXNDC11(M1A) cDNA tagged with Flag at the C-terminus, which can produce a soluble protein, and TAP-EDEM2 cDNA, and were able to extract sufficient EDEM2-TXNDC11 complex after lysis for subsequent purification (*Figure 7*). Our results represent the first clear demonstration of in vitro mannosidase activity of EDEM family proteins toward free oligosaccharides, an effect which is much clearer than in the case of yeast Htm1p.

In conclusion, the notion that EDEM2 catalyzes the first mannose trimming step and thereby initiates gpERAD is now firmly supported by our genetic and biochemical analyses.

# Materials and methods

**Key resources table**

| Reagent type or resources | Designation | Source or reference | Identifier | Additional information |
|---|---|---|---|---|
| Cell line (*Homo sapiens*) | colorectal carcinoma | ATCC | HCT116 | Parental HCT116 cell line have been authenticated and all cell lines have been tested negative for mycoplasma. |
| Recombinant DNA reagent | p3xFlag-CMV-14 | Sigma-Aldrich | | |
| Recombinant DNA reagent | pcDNA3.1-MycHis | ThermoFisher | | |
| Antibody | anti-TXNDC11 (Rabbit monoclonal) | Abcam | Cat#: ab188329 | WB (1:500) |
| Antibody | anti-EDEM2 (Rabbit polyclonal) | Novusbio | Cat#: NBP2-37921 | WB (1:500) |
| Antibody | anti-HA (Rabbit polyclonal) | Recenttec | Cat#: R4-TP1411100 | WB (1:1000) |
| Antibody | anti-calnexin (Rabbit polyclonal) | Enzo Life Sciences | Cat#: ADI-SPA-865 | WB (1:1000) |
| Antibody | anti-PDI (Rabbit polyclonal) | Enzo Life Sciences | Cat#: ADI-SPA-890 | WB (1:1000) |
| Antibody | anti-ERp72 (Rabbit polyclonal) | Enzo Life Sciences | Cat#: ADI-SPA-720 | WB (1:1000) |
| Antibody | anti-calreticulin (Rabbit polyclonal) | Enzo Life Sciences | Cat#: ADI-SPA-600 | WB (1:1000) |
| Antibody | anti-Flag (Mouse monoclonal) | Sigma | Cat#: F3165 | WB (1:1000) IP (2.5 µl) |
| Antibody | anti-β-actin (Mouse monoclonal) | Wako | Cat#: 017–24573 | WB (1:2000) |
| Antibody | Anti-human ATF6α (Rabbit polyclonal) | *Haze et al., 1999* | | WB (1:1000) |

## Statistics

Statistical analysis was conducted using Student's t-test, with probability expressed as *p<0.05 and **p<0.01 for all figures.

## Construction of plasmids

Recombinant DNA techniques were performed according to standard procedures (*Sambrook et al., 1989*) and the integrity of all constructed plasmids was confirmed by extensive sequencing analyses. Site-directed mutagenesis was carried out using DpnI. A p3xFlag-CMV-14 expression vector (Sigma) and pcDNA3.1-MycHis expression vector (ThermoFisher) were used to express a protein (EDEM2 or TXNDC11) tagged with 3xFlag and c-Myc at the C-terminus, respectively. pCMV-SP-TAP-EDEM2 was constructed based on pCMV, pcDNA3.1-SP-2xProA-2xTEV-6xMyc and pCMV-EDEM2-3xFlag using an NEBuilder HiFi DNA Assembly Cloning Kit (New England Biolabs). SP denotes a signal peptide. The ERAD-Ls substrate mCD3-δ-ΔTM-HA was the kind gift of Maurizio Molinari at the Institute for Research in Biomedicine, Switzerland.

## Cell culture, transfection and N-glycan profiling

HCT116 cells (ATCC CCL-247) were cultured in Dulbecco's modified Eagle's medium (glucose 4.5 g/liter) supplemented with 10% fetal bovine serum, 2 mM glutamine, and antibiotics (100 U/ml penicillin and 100 μg/ml streptomycin) at 37°C in a humidified 5% CO2/95% air atmosphere. Transfection was performed using mainly polyethylenimine max (Polyscience) and partly Lipofectamine 2000 (Invitrogen) according to the manufacturers' instructions. Pyridylamination and structural identification of N-glycans of total cellular glycoproteins were performed as described previously (*Horimoto et al., 2013*; *Ninagawa et al., 2014*). EndoH was obtained from Calbiochem; cycloheximide from Sigma; MG132 from Peptide Institute; and trypsin and protease inhibitor cocktail from Nacalai Tesque. Trypsin digestion of cell lysates was carried out as described previously (*Ninagawa et al., 2015*).

## Immunological techniques

Immunoblotting analysis was carried out according to the standard procedure (*Sambrook et al., 1989*) as described previously (*Ninagawa et al., 2011*). Chemiluminescence obtained using Western Blotting Luminol Reagent (Santa Cruz Biotechnology) was detected using an LAS-3000mini Lumino-Image analyzer (Fuji Film). Rabbit monoclonal anti-TXNDC11 antibody was obtained from Abcam. Rabbit polyclonal anti-EDEM2 antibody was obtained from Novusbio; and anti-HA from Recenttec. Rabbit polyclonal anti-calnexin, anti-PDI, and anti-ERp72, and anti-calreticulin antibodies were obtained from Enzo Life Sciences. Mouse monoclonal anti-Flag antibody was obtained from Sigma; and anti-β-actin from Wako. Anti-human ATF6α antibody was produced previously (*Haze et al., 1999*).

Immunoprecipitation was performed using anti-Flag antibody and protein G-coupled Sepharose beads (GE Healthcare). Beads were washed with high salt buffer (50 mM Tris/Cl, pH 8.0, containing 1% NP-40 and 150 mM NaCl) twice, washed with PBS, and boiled in Laemmli's sample buffer.

## CRISPR/Cas9-based PITCh system

*TXNDC11* gene knockout was carried out as previously described (*Sakuma et al., 2016*). Briefly, the oligonucleotides 5'-caccGCAGCGCGCAGCCGAGCGCCA-3' and 5'-aaacTGGCGCTCGGCTGCGCGCTGC-3' to express gRNA for cleavage of exon 1 of the *TXNDC11* gene (lower case letters denote cohesive ends for ligation) were annealed and inserted into the BpiI site of pX330A-1x2 (Addgene). The BsaI fragment of pX330S-2-PITCh to express PITCh gRNA (Addgene) was inserted into the resulting vector to create pX330A-TXNDC11-PITCh1x2 (*Figure 3—figure supplement 1A*). The 5' forward primer 5'-ccgcgttacatagcatcgtacgcgtacgtgtttggAGCTGCTCTGCGGGGCCGTG-ccggatccggcccgctagcacgtattta-3' corresponding to the PITCh gRNA target site (5'-side lower case letters), the left arm of the *TXNDC11* gene (upper case letters) and the vector sequence (3'-side lower case letters), as well as the 3' reverse primer 5'-acgcgtacgtgtttggAGGAGCAGCGCGCAGCC-GAGcgcgcccttaagtcgacaag-3' corresponding to the PITCh gRNA target site (5'-side lower case letters), the right arm of the *TXNDC11* gene (upper case letters) and the vector sequence (3'-side lower case letters) were used to amplify the region containing the puromycin-resistant gene but not the EGFP gene in pCRIS-PITChv2-FBL followed by infusion reaction to create pCRIS-Puro (*Figure 3—*

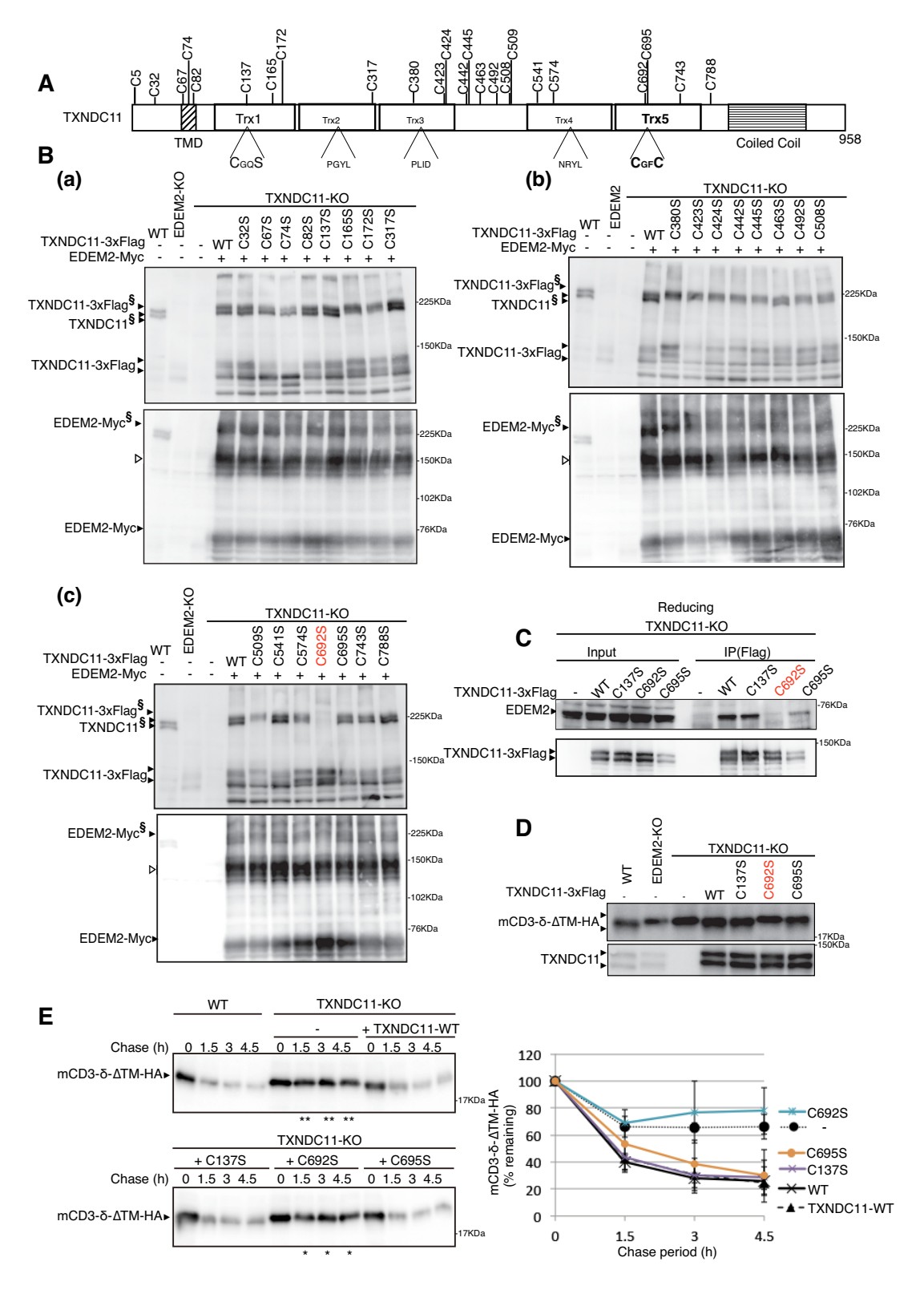

**Figure 6.** Effect of mutation of various cysteine residues of TXNDC11 on its disulfide-bonding to EDEM2 and on gpERAD. (**A**) Structure of human TXNDC11 is schematically shown with cysteine residues (C) highlighted together with their positions. (**B**) Cell lysates were prepared from WT, EDEM2-KO, and TXNDC11-KO cells expressing WT or one of various cysteine mutants of 3x Flag-tagged TXNDC11 together with Myc-tagged EDEM2 by transfection, subjected to SDS-PAGE under non-reducing conditions, and analyzed by immunoblotting using anti-TXNDC11 and anti-EDEM2

*Figure 6 continued on next page*

*Figure 6 continued*

antibodies. (C) Cell lysates were prepared from TXNDC11-KO cells expressing WT or one of the three cysteine mutants of 3x Flag-tagged TXNDC11, and subjected to immunoprecipitation using anti-Flag antibody. An aliquot of cell lysates (Input) and the immunoprecipitates {IP(Flag)} were subjected to SDS-PAGE under reducing conditions, and analyzed by immunoblotting using anti-EDEM2 and anti-TXNDC11 antibodies. (D) Cell lysates were prepared from WT, EDEM2-KO, and TXNDC11-KO cells expressing WT or one of the three cysteine mutants of 3x Flag-tagged TXNDC11 together with mCD3-δ-ΔTM-HA by transfection, and analyzed by immunoblotting using anti-HA and anti-TXNDC11 antibodies. (E) Cycloheximide chase was conducted to determine the degradation rate of mCD3-δ-ΔTM-HA in WT and TXNDC11-KO cells expressing WT or one of the three cysteine mutants of 3x Flag-tagged TXNDC11 by transfection, and cell lysates were analyzed by immunoblotting using anti-HA antibody (n = 3). Quantified data are shown on the right.

figure supplement 1A). HCT116 cells were transfected with pX330A-TXNDC11-PITCh1x2 and pCRIS-Puro using polyethylenimine max and screened for puromycin resistance.

## Genomic PCR

Homologous recombination in HCT116 cells was confirmed by genomic PCR using a pair of primers 753Fw and 1708Rv, 730Fw and 1831Rv, and TgFw and TgRv.

## RT-PCR

Total RNA prepared from cultured cells (~5 × 10^6 cells) by the acid guanidinium/phenol/chloroform method using ISOGEN (Nippon Gene) was converted to cDNA using Moloney murine leukemia virus reverse transcription (Invitrogen) and random primers. The full-length open reading frame of TXNDC11 and GM130 was amplified using PrimeSTAR HS DNA polymerase (Takara Bio) and a pair of primers, TXNDC11-Fw and TXNDC11-Rv, and GM130-Fw and GM130-Rv, respectively.

## Quantitative RT-PCR

Total RNA extracted as above was subjected to quantitative RT-PCR analysis using the SYBR Green method (Applied Biosystems) and a pair of primers, namely 778Fw and 838Rv as well as 1036Fw and 1105Rv for *EDEM2* mRNA, XBP1-Fw and XBP1-Rv: for spliced *XBP1* mRNA, and GAPDH-Fw and GAPDH-Rv for *GAPDH* mRNA.

## Repeated freezing and thawing

TXNDC11-KO cells in 10 cm dishes were collected 16 hr after transfection in 1 ml of PBS containing protease inhibitor cocktail and MG132, and centrifuged at 5,000 rpm for 3 min. The pellets were resuspended in 600 μl of PBS containing protease inhibitor cocktail and MG132, quickly frozen in liquid nitrogen, and then quickly thawed in a water bath at 37°C. This freezing and thawing cycle was repeated 10 times. After centrifugation at 1000 x g for 10 min, the supernatant was collected, subjected to further 5 cycles of freezing and thawing, and centrifuged at 1000 x g for 10 min. Half of the supernatant was used as total membrane fraction (T). The second half of the supernatant was centrifuged at 40,000 rpm for 30 min at 4°C to collect supernatant (S) and pellets (P).

## Purification of EDEM2

EDEM2-KO cells plated on 15 cm dishes were simultaneously transfected with TAP-EDEM2 and TXNDC11(M1A)−3xFlag. Forty eight hours later the cells were lysed in lysis buffer {50 mM MES, pH 7.5, containing 150 mM NaCl, 1% CHAPS, and EDTA-free protease inhibitor cocktail (Roche)}, and centrifuged at 9,500 g for 30 min at 4°C. The resulting supernatant was filtrated through a low protein binding syringe filter (Merck) and rotated for 8 hr at 4°C after the addition of IgG Sepharose beads (GE Health Care). The beads were collected by centrifugation at 3,000 rpm for 1 min at 4°C, washed twice briefly and then washed overnight with wash buffer (50 mM MES, pH7.5, containing 400 mM NaCl, 0.1% CHAPS and EDTA-free protease inhibitor cocktail). The beads were incubated with 200 U of AcTEV protease (Invitrogen) in TEV buffer (50 mM MES, pH7.5, containing 150 mM NaCl) for 24 hr at 4°C, and then centrifuged briefly. The resulting supernatant was concentrated using an Amicon Filter (10 kDa cut off, Millipore) by centrifugation at 4,000 g for 1 hr at 4°C. During concentration the buffer was changed to 50 mM MES, pH7.5, containing 150 mM NaCl and 5 mM CaCl$_2$, by three additions to the filter.

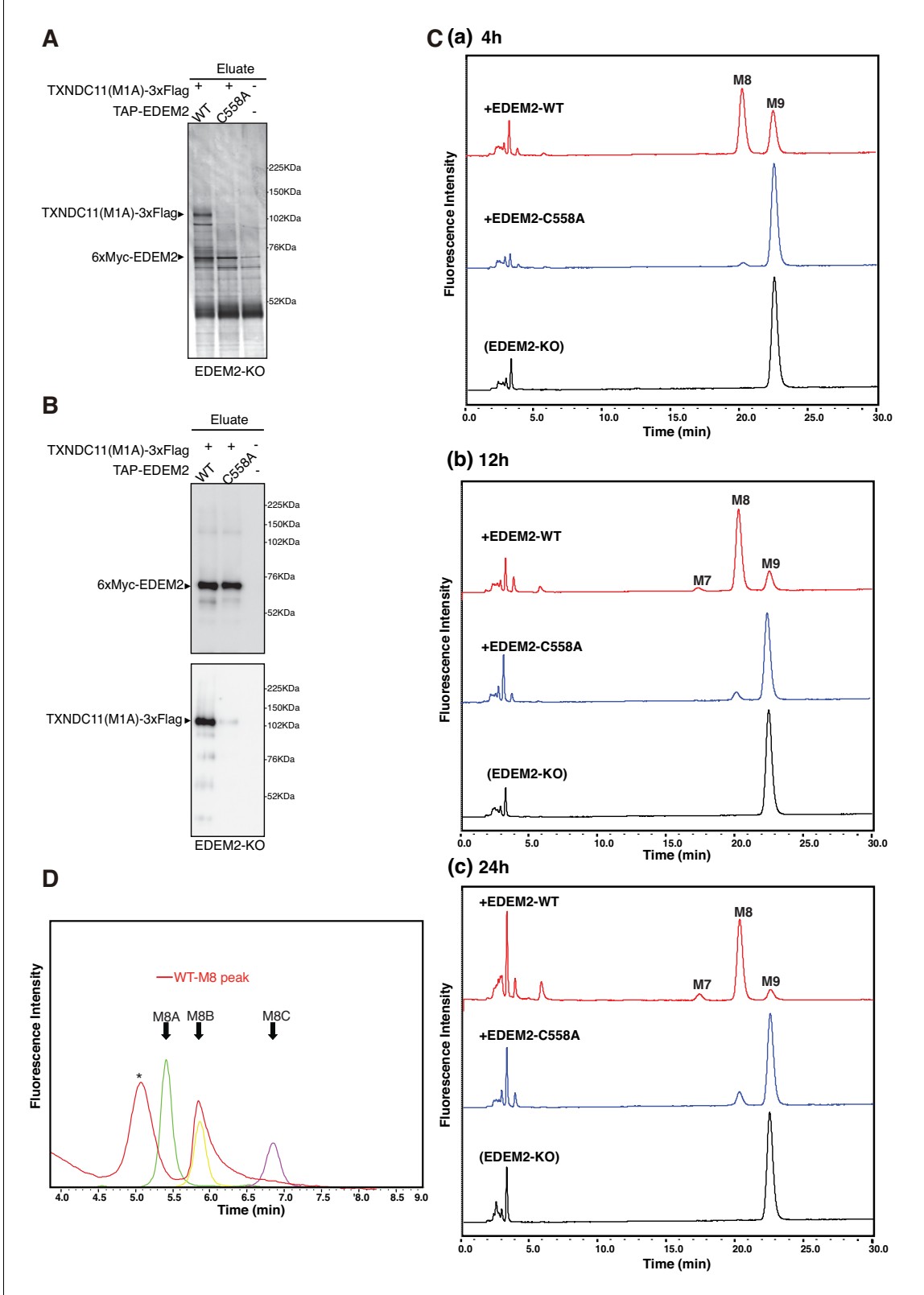

**Figure 7.** Mannosidase activity of the EDEM2-TXNDC11 complex. (**A**) EDEM2-KO cells were untransfected (-) or transfected (+) with the indicated plasmids, and subjected to purification using IgG Sepharose beads. Samples eluted from the beads by digestion with TEV protease were subjected to reducing SDS-PAGE and silver-staining. (**B**) The amounts of 6xMyc-EDEM2 and TXNDC11(M1A)-3xFlag in samples for in vitro assay were checked by immunoblotting using anti-EDEM2 and anti-TXNDC11 antibodies. (**C**) PA-M9 was incubated with samples in (**B**) for 4 hr (**a**), 12 hr (**b**) and 24 hr (**c**) as

*Figure 7 continued on next page*

*Figure 7 continued*

indicated, and then analyzed by HPLC (amide column) for mannose contents. This experiment was completed once. (D) The M8 peak obtained in (C) after incubation with WT EDEM2 was analyzed by HPLC (ODS column) for isomer identification. Green, yellow, and purple peak indicate the elution position of M8A, M8B, and M8C, respectively. The asterisk observed during analysis of the M8 peak denotes a fluorescent peak unrelated to oligosaccharides. This experiment was completed once.

## In vitro mannosidase assay

PA-labeled free oligosaccharides were purchased from Takara Bio. Approximately 90 ng of purified WT and C558A mutant of EDEM2 were incubated with 50 µM PA-M9 in a total volume of 45 µl of Assay buffer (50 mM MES, pH 7.5, containing 150 mM NaCl and 5 mM $CaCl_2$) for 4 hr, 12 hr, or 24 hr at 37°C. The reaction was stopped by boiling for 5 min. The samples were evaporated, dissolved with 20 µl of 70% (v/v) acetonitrile, and analyzed using a TSK-gel Amide-80 column (Tosoh) for mannose contents. Identification of *N*-glycan structures was based on their elution positions on the column and their molecular mass values compared with those of PA-glycans in the GALAXY database (http://www.glycoanalysis.info/galaxy2/ENG/index.jsp) (*Takahashi and Kato, 2003*). The M8 peak from the WT sample was collected, evaporated, dissolved with 20 µl of water, and analyzed using a Shim-pack HRC-octadecyl silica column (Shimadzu) for isomer identification.

## Acknowledgements

The authors declare no competing financial interests. We are grateful to Dr. Masao Sakaguchi at Hyogo Prefectural University for helpful advice on signal peptide cleavage. We thank Ms. Kaoru Miyagawa for her technical and secretarial assistance. This work was financially supported in part by grants from MEXT, Japan (18K06216 to S N, 17H06414 to H Y, 19K06658 to T I, 18K06110 to T O, 17H01432 and 17H06419 to K M) and by the Joint Research by Exploratory Research Center on Life and Living Systems (19–303 to K M).

## Additional information

### Funding

| Funder | Grant reference number | Author |
| --- | --- | --- |
| Ministry of Education, Culture, Sports, Science, and Technology | 18K06216 | Satoshi Ninagawa |
| Ministry of Education, Culture, Sports, Science, and Technology | 17H06414 | Hirokazu Yagi |
| Ministry of Education, Culture, Sports, Science, and Technology | 19K06658 | Tokiro Ishikawa |
| Ministry of Education, Culture, Sports, Science, and Technology | 18K06110 | Tetsuya Okada |
| Ministry of Education, Culture, Sports, Science, and Technology | 17H01432 | Kazutoshi Mori |
| Ministry of Education, Culture, Sports, Science, and Technology | 17H06419 | Kazutoshi Mori |

The funders had no role in study design, data collection and interpretation, or the decision to submit the work for publication.

## Author contributions
Ginto George, Satoshi Ninagawa, Hirokazu Yagi, Taiki Saito, Data curation, Formal analysis, Investigation; Tokiro Ishikawa, Investigation; Tetsushi Sakuma, Takashi Yamamoto, Koshi Imami, Yasushi Ishihama, Koichi Kato, Methodology; Tetsuya Okada, Supervision; Kazutoshi Mori, Conceptualization, Supervision, Funding acquisition

## Author ORCIDs
Ginto George https://orcid.org/0000-0003-4804-9594
Satoshi Ninagawa https://orcid.org/0000-0002-8005-4716
Hirokazu Yagi http://orcid.org/0000-0001-9296-0225
Taiki Saito https://orcid.org/0000-0002-7580-4340
Tokiro Ishikawa http://orcid.org/0000-0003-1718-6764
Tetsushi Sakuma https://orcid.org/0000-0003-0396-1563
Koshi Imami https://orcid.org/0000-0002-7451-4982
Yasushi Ishihama https://orcid.org/0000-0001-7714-203X
Koichi Kato https://orcid.org/0000-0001-7187-9612
Tetsuya Okada https://orcid.org/0000-0002-2513-1301
Kazutoshi Mori https://orcid.org/0000-0001-7378-4019

## Decision letter and Author response
Decision letter https://doi.org/10.7554/eLife.53455.sa1
Author response https://doi.org/10.7554/eLife.53455.sa2

# Additional files

## Supplementary files
• Supplementary file 1. Table of primers.
• Transparent reporting form

## Data availability
All data generated or analysed during this study are included in the manuscript.

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
