## [Decision Letter]

**Acceptance summary:**

As a crucial part of quality control in the ER it has been believed that Man9 to Man8B conversion on N-glycans is carried out by the action of "ER α-mannosidase" i.e. Man1B1. The authors, on the other hand, previously showed, using various gene-KO cell lines, that it is actually most likely EDEM2 that is responsible. There are, however, conflicting reports and especially the evidence for the in vitro activity of EDEM2 has so far been lacking.

In this study, the authors convincingly showed that (1) EDEM2 form a complex with TXNDC11 through C558 of EDEM2 and C692 of TXNDC11; (2) the interaction is important for the α-mannosidase activity of EDEM2, and (3) the purified EDEM2-TXNDC11 complex can exert Man9-to-Man8B conversion activity in vitro.

While the interaction of EDEM2 and TXNDC11 had been demonstrated, its biological significance was unclear. Overall, this study makes a big step forward in our understanding of the molecular mechanism of glycoprotein ERAD. The experiments are carefully designed, analyses are thorough, and the conclusion is convincing.

**Decision letter after peer review:**

Thank you for submitting your article "EDEM2 Stably Disulfide-bonded to TXNDC11 Catalyzes the First Mannose Trimming Step in Mammalian Glycoprotein ERAD" for consideration by *eLife*. Your article has been reviewed by three peer reviewers, one of whom is a member of our Board of Reviewing Editors, and the evaluation has been overseen by Vivek Malhotra as the Senior Editor. The following individuals involved in review of your submission have agreed to reveal their identity: Tadashi Suzuki (Reviewer #2); Daniel N Hebert (Reviewer #3).

The reviewers have discussed the reviews with one another and the Reviewing Editor has drafted this decision to help you prepare a revised submission. The experiments are carefully designed, analyses are thorough, and the conclusion is convincing. All issues raised are considered minor and can be addressed by revisions to the text.

Minor points:

Please see the minor points below to be addressed in the revision. The issue of glycoform separation (raised by reviewer #3) was briefly discussed by the reviewers and may not be an issue. Nevertheless, the text may better clarify the extent of resolution in the assay.

Reviewer #2:

1) Introduction, first paragraph: the triplet code (Asn-X-Ser/Thr); to be precise, it can be mentioned as (Asn-X-Ser/Thr; X: any amino acids except Pro).

2) Subsection “Immunological techniques” – it will be of great help if the authors specify catalogue numbers, together with the vendors, for the origin of antibodies used.

3) Figure 5A/B; these asterisks, I assume, shows the statistical significance, just like Figure 1E (*P<0.05, *P<0.01), but it may be of help if the authors also mention this in Figure 5A/B legends. And what exactly is the method for the statistical testing?

4) Figure 5G; The authors explanation that M1A protein becomes soluble by using transmembrane domain as a signal peptide sounds reasonable; do the authors actually know that the transmembrane domain is cleaved off (to become a soluble form)? This is not an absolute requirement for this manuscript, but I am just curious.

5) Figure 6C/D; in C692S expressing cells, the size of TXBDC11 looks similar with others, despite the (seemingly) lack of EDEM2 activity. Is this just a specific for this particular construct (i.e. somehow the migration position becomes indistinguishable because of the slight change of the mobility of this particular mutant – if that is the case after Endo H digestion the mobility may be a bit different between WT and C692S mutant)? I sort of expected that there may be a shift on mobility, just like the one expressing in EDEM2 KO cell s (ex. Figure 2A (c)).

Reviewer #3:

It is an overstatement to say that "results clearly showed that Man9GlcNAc2 (M9 hereafter) is converted to Man8GlcNAc2[…] by EDEM2" in the Introduction as the previous study in Ninagawa et al., 2014 (Figure 1D, Supplementary Figure 2A and Supplementary 3F) that the statement is referring to did not appear to allow separation of the various Man8 glycoforms (A, B and C). While quants are included for the different glycoforms, the HPLC chromatogram does not appear to support adequate separation of the glycoforms. Please explain. A similar statement is also found in the Discussion section. This overstatement also undercuts the importance of the present in vitro demonstration and the adequate separation of Man8A, B and C PA-tagged glycoforms now demonstrated in the current Figure 7D.

Figure 3 should be a supplementary figure.

Figure 4D requires quantification to support the statement in the second paragraph of the subsection “TXNDC11 knockout eliminates high molecular weight forms of EDEM2 required for gpERAD”.

The explanation that slight changes to culture conditions altered the glycoforms accumulated in the EDEM2-KO cells previously observed (Ningawa et al., 2014) compared to the present study seems unlikely. Was the same carbohydrate analysis approach used to identify the glycoforms accumulated in both studies?

Interestingly, TXNDC11 appears to have two start methionines. M1A forces the use of the downstream Met (M58) as a translational start and produces a product found to be 50% in the pellet and 50% in the supernatant after alkaline extraction. This is concluded to mean the protein is soluble (subsection “TXNDC11 knockout eliminates high molecular weight forms of EDEM2 required for gpERAD”, last paragraph). Why is it not that 50% is soluble and 50% is luminal as previously found for EDEM1 (Tamura et al., 2011)? Only a membrane protein control is shown (calnexin) for the alkaline extraction. A soluble protein control should also be shown.

The analogous yeast study described (Gauss et al., 2011) found that Htm1p did not act as a mannosidase without its associated oxidoreductase Pdi1p. It should also be pointed out that this very weak demonstration of mannosidase activity required the glycan to be attached to a protein (no mannosidase on free N-glycans); whereas, in the current study the mannosidase activity of EDEM2 is more robust and demonstrated using PA-tagged glycans (Note: PA needs to be defined in the subsection “in vitro mannosidase assay”).

---

## [Author Response]

Minor points:Please see the minor points below to be addressed in the revision. The issue of glycoform separation (raised by reviewer #3) was briefly discussed by the reviewers and may not be an issue. Nevertheless, the text may better clarify the extent of resolution in the assay.Reviewer #2:1) Introduction, first paragraph: the triplet code (Asn-X-Ser/Thr); to be precise, it can be mentioned as (Asn-X-Ser/Thr; X: any amino acids except Pro).

We have done so (Introduction, first paragraph).

2) Subsection “Immunological techniques” – it will be of great help if the authors specify catalogue numbers, together with the vendors, for the origin of antibodies used.

We have added such information (Materials and methods).

3) Figure 5A/B; these asterisks, I assume, shows the statistical significance, just like Figure 1E (*P<0.05, *P<0.01), but it may be of help if the authors also mention this in Figure 5A/B legends. And what exactly is the method for the statistical testing?

We have added“Statistics:Statistical analysis was conducted using Student's t-test, with probability expressed as *p<0.05 and **p<0.01 for all figures” at the top of Materials and methods.

4) Figure 5G; The authors explanation that M1A protein becomes soluble by using transmembrane domain as a signal peptide sounds reasonable; do the authors actually know that the transmembrane domain is cleaved off (to become a soluble form)? This is not an absolute requirement for this manuscript, but I am just curious.

We have conducted additional experiments (Figures 5D, 5E and 5F) and have described that “Centrifugal fractionation after repeated freezing and thawing of cells indicated that the M58A mutant was a membrane protein like calnexin, whereas the M1A mutant was split into a calnexin-like membrane protein and a calreticulin-like soluble protein (Figure 5C). […] We concluded that M1A becomes a soluble protein if it is cleaved by signal peptidase and that M1A uncleaved by chance remains as a M58A-like membrane protein.”

5) Figure 6C/D; in C692S expressing cells, the size of TXBDC11 looks similar with others, despite the (seemingly) lack of EDEM2 activity. Is this just a specific for this particular construct (i.e. somehow the migration position becomes indistinguishable because of the slight change of the mobility of this particular mutant – if that is the case after Endo H digestion the mobility may be a bit different between WT and C692S mutant)? I sort of expected that there may be a shift on mobility, just like the one expressing in EDEM2 KO cell s (ex. Figure 2A (c)).

It is probably due to transfection-mediated overexpression. The migration position of even WT TXNDC11-3xFlag did not differ between WT and EDEM2-KO cells (please see below). We consider that endogenous EDEM2 was not sufficient for full conversion from M9 to M8 present on WT TXNDC11-3xFlag. Therefore, after co-transfection with EDEM2-Myc, WT but not C692S TXNDC11-3xFlag migrated slightly faster.

Reviewer #3:It is an overstatement to say that "results clearly showed that Man9GlcNAc2 (M9 hereafter) is converted to Man8GlcNAc2[…] by EDEM2" in the Introduction as the previous study in Ninagawa et al., 2014 (Figure 1D, Supplementary Figure 2A and Supplementary Figure 3F) that the statement is referring to did not appear to allow separation of the various Man8 glycoforms (A, B and C). While quants are included for the different glycoforms, the HPLC chromatogram does not appear to support adequate separation of the glycoforms. Please explain. A similar statement is also found in the Discussion section. This overstatement also undercuts the importance of the present in vitro demonstration and the adequate separation of Man8A, B and C PA-tagged glycoforms now demonstrated in the current Figure 7D.

Please see Figure S3, panel F from Ninagawa et al., 2014, also Materials and methods in Ninagawa et al., 2014: “N-glycan structures were identified based on their elution times from these columns in comparison with those of PA-glycans in the GALAXY database (Takahashi and Kato, 2003) and confirmed by co-chromatography with standard PA-high mannose type oligosaccharides (Kamiya et al., 2008; Tomiya et al., 1991).”

As our JCB paper in 2014 was published as a report, we did not publish all detailed data. Instead, we briefly described as above. Each peak fraction was collected and analyzed by co-chromatography with standard PA-high mannose type oligosaccharides, just like the experiment shown in Figure 7D, allowing us to quantify each glycoform.

The statements were changed to “is required for”.

Figure 3 should be a supplementary figure.

We have done so.

Figure 4D requires quantification to support the statement in the second paragraph of the subsection “TXNDC11 knockout eliminates high molecular weight forms of EDEM2 required for gpERAD”.

We have done so (Figure 3D).

The explanation that slight changes to culture conditions altered the glycoforms accumulated in the EDEM2-KO cells previously observed (Ninagawa et al., 2014) compared to the present study seems unlikely. Was the same carbohydrate analysis approach used to identify the glycoforms accumulated in both studies?

Yes, it was. We have described that “accumulation of Glc_1_Man_9_GlcNAc_2_ (GM9) was probably due to a subtle change in culture conditions, because GM9 was not accumulated in EDEM2-KO human HCT116 cells in our previous study (Ninagawa et al., 2014). Alternatively, it may be due to enhanced cellular re-glucosylation activity caused by blockage of mannose trimming from M9 to M8 (Molinari, 2007), because both GM9 and M9 were accumulated in EDEM2-KO chicken DT40 cells in our previous study (Ninagawa et al., 2014)”. See Supplementary Figure 2 from Ninagawa et al., 2014.

Interestingly, TXNDC11 appears to have two start methionines. M1A forces the use of the downstream Met (M58) as a translational start and produces a product found to be 50% in the pellet and 50% in the supernatant after alkaline extraction. This is concluded to mean the protein is soluble (subsection “TXNDC11 knockout eliminates high molecular weight forms of EDEM2 required for gpERAD”, last paragraph). Why is it not that 50% is soluble and 50% is luminal as previously found for EDEM1 (Tamura et al., 2011)? Only a membrane protein control is shown (calnexin) for the alkaline extraction. A soluble protein control should also be shown.

Please refer to our response to the comment 4 of the reviewer 2.

The analogous yeast study described (Gauss et al., 2011) found that Htm1p did not act as a mannosidase without its associated oxidoreductase Pdi1p. It should also be pointed out that this very weak demonstration of mannosidase activity required the glycan to be attached to a protein (no mannosidase on free N-glycans); whereas, in the current study the mannosidase activity of EDEM2 is more robust and demonstrated using PA-tagged glycans (Note: PA needs to be defined in the subsection “in vitro mannosidase assay”).

We have changed to “Importantly, Htm1p-Pdi1p complex purified from insect cells, in which glutathione S-transferase-Htm1p fusion protein and His-tagged Pdi1p are simultaneously expressed, converted approximately 10% of M8 to M7 in vitro when M8 is attached to unfolded, reduced and alkylated polypeptides; in contrast, Htm1p-Pdi1p complex was nearly inactive against free M8 (Gauss et al., 2011).”.

We have added “Our results represent the first clear demonstration of in vitro mannosidase activity of EDEM family proteins toward free oligosaccharides, an effect which is much clearer than in the case of yeast Htm1p.”

PA is defined as pyridylamine in the Introduction.